# A Parametric Analysis of Capillary Height in Single-Layer, Small-Scale Microfluidic Artificial Lungs

**DOI:** 10.3390/mi13060822

**Published:** 2022-05-25

**Authors:** Lindsay J. Ma, Emmanuel A. Akor, Alex J. Thompson, Joseph A. Potkay

**Affiliations:** 1Department of Surgery, University of Michigan, Ann Arbor, MI 48109, USA; lindsma@med.umich.edu (L.J.M.); eaakor@umich.edu (E.A.A.); ajthomp@umich.edu (A.J.T.); 2Veterans Affairs Ann Arbor Healthcare System, Ann Arbor, MI 48109, USA

**Keywords:** analytical models, biomedical engineering, biomedical materials, biomembranes, CFD simulations, fluidic microsystems, mathematical model, microfabrication, microfluidics, polymer films

## Abstract

Microfluidic artificial lungs (μALs) are being investigated for their ability to closely mimic the size scale and cellular environment of natural lungs. Researchers have developed μALs with small artificial capillary diameters (10–50 µm; to increase gas exchange efficiency) and with large capillary diameters (~100 µm; to simplify design and construction). However, no study has directly investigated the impact of capillary height on μAL properties. Here, we use Murray’s law and the Hagen-Poiseuille equation to design single-layer, small-scale μALs with capillary heights between 10 and 100 µm. Each µAL contained two blood channel types: capillaries for gas exchange; and distribution channels for delivering blood to/from capillaries. Three designs with capillary heights of 30, 60, and 100 µm were chosen for further modeling, implementation and testing with blood. Flow simulations were used to validate and ensure equal pressures. Designs were fabricated using soft lithography. Gas exchange and pressure drop were tested using whole bovine blood. All three designs exhibited similar pressure drops and gas exchange; however, the μAL with 60 µm tall capillaries had a significantly higher wall shear rate (although physiologic), smaller priming volume and smaller total blood contacting surface area than the 30 and 100 µm designs. Future μAL designs may need to consider the impact of capillary height when optimizing performance.

## 1. Introduction

Hollow fiber membrane (HFM) lungs are used in millions of cardiopulmonary bypass procedures each year and provide lung support to tens of thousands of patients with lung disease [1,2]. Recent advances have also made ambulatory HFM lung use possible within intensive care units [3] and occasionally in emergency departments [4]. Despite the many applications for HFM lungs, these devices subject blood to non-physiologic flow patterns, including areas of stagnation and high shear, as well as large artificial surface areas, which predispose blood to clotting [5]. Systemic anticoagulation is therefore necessary to mitigate platelet activation and thrombus formation [6,7] but simultaneously increases hemorrhagic complication risks [8]. Furthermore, HFM lungs are limited in their gas exchange efficiency due to the large gas diffusion distances inherent to the hollow fiber mats from which HFMs are formed [5]. Despite utilizing large surface areas and 100% oxygen sweep gases, HFM lungs can only support the gas exchange demands of patients at rest [9].

To overcome the drawbacks of HFM lungs, researchers have investigated using microfluidic technology to develop artificial lungs that more closely mimic their natural counterpart. These microfluidic artificial lungs (μALs) can exhibit µm-tall artificial capillaries, thereby increasing gas exchange efficiency and achieving physiologic blood flow paths, potentially increasing biocompatibility compared to current alternatives. To date, our and other research groups have demonstrated small-scale, single-layer two-dimensional (2D) µALs with record gas exchange efficiency [10,11], biomimetic blood flow networks [10,11,12,13,14,15,16,17,18], surface coatings to reduce protein and platelet deposition and increase lifetime [15,19,20,21,22], and have also shown that endothelial cells can be grown in microfluidic blood flow networks and significantly reduce thrombus area [16,23,24,25]. Manufacturing techniques have been demonstrated to increase the blood flow capacity of µAls and move these devices towards clinical application [11,14,16,26,27,28,29,30,31,32]. These previous µALs have been designed with small artificial capillary diameters (10–40 μm) to maximize gas exchange efficiency or with larger diameters (~100 μm) to simplify construction and potentially minimize clotting. However, no study to date has directly compared the impact of capillary lumen size on μAL properties and performance. This study seeks to fill that gap in knowledge.

In this work, μAL designs with capillary heights (H_c_) between 10 and 100 μm are compared for a fixed set of performance specifications (rated blood flow and blood-side pressure drop). All designs contained blood distribution channels and artificial capillaries and were constrained to fit on a 6″ silicon wafer. Rated blood flow and total pressure drop were fixed at 0.8 mL/min and 50 mmHg for all designs. Surface area (gas exchange and blood contacting), blood priming volume in capillaries and distribution channels (V_prime_) capillary wall shear rate and number of bifurcations were determined for each capillary height and compared. Blood flow was modeled in three designs (H_c_ = 30, 60, 100 μm) using CFD, implemented via soft lithography and tested in vitro with bovine blood to verify performance. A preliminary version of this work was accepted as a conference abstract and presented at the ASAIO 66th Annual Conference in June 2021 [33].

## 2. Materials and Methods

### 2.1. Materials

Channel dimensions and angles were calculated using Microsoft Excel (Redmond, WA, USA). 2-dimensional (2D) blood flow paths were designed in DraftSight (Dassault Systemes, Waltham, MA, USA). SolidWorks (Dassault Systemes, Concord, MA, USA) was used to render 3-dimensional (3D) designs and run flow simulations (CFD). Silicon wafers were purchased from University Wafer (Boston, MA, USA) and Microchem SU-8 2035 and SU-8 2100 permanent epoxy negative photoresist were purchased from Kayaku Advanced Materials (Westborough, MA, USA). Custom designed photomasks (10,160 DPI laser plotted) were purchased from FineLine Imaging (Colorado Springs, CO, USA). Acetone, 2-isopropanol and SU-8 developer were purchased from Fisher Scientific (Pittsburg, PA, USA). Dow Corning 3140 RTV silicone conformal coating and Dow Corning Sylgard 184 (PDMS) silicone elastomer base with curing agent were obtained from Ellsworth Adhesives (Germantown, WI, USA). A Dino-Scope 2.0 (Dino-Lite, Torrance, CA, USA) was used for optical imaging. A MARCH AP-300 Plasma System (Nordson, Carlsbad, CA, USA) was used for plasma activation of PDMS surfaces for bonding. Silastic tubing, luer connectors, three-way stop cocks and Masterflex Digital Benchtop Gear Pump System were purchased from Cole Parmer (Vernon Hills, IL, USA). An i-STAT Handheld blood analyzer, ACTc cartridges and CG8+ cartridges were obtained from Abbott Point of Care (Princeton, NJ, USA). Bovine whole blood was obtained from Lampire Biological Laboratories (Pipersville, PA, USA). Pressure measurements were achieved using a Hewlett Packard M1094B pressure monitor (Houston, TX, USA), Philips Medizin Systeme M1006B (Böblingen, Germany), Transpac ICU Medical Hospira Reusable Cables (San Clemente, CA, USA) and Transpac disposable pressure transducers (San Clemente, CA, USA).

### 2.2. Mathematical Models of μALs with 10–100 μm Tall Capillaries

In this study, all μALs were designed to contain a blood flow layer, a non-porous gas-diffusion PDMS membrane and a gas flow layer (Figure 1). The Hagen-Poiseuille equation, a previously developed model of gas exchange in μALs [34], and Murray’s law were used to compare μAL designs with capillary heights between 10 and 100 μm, as described below [34,35,36,37,38]. Goal parameters were rated blood flows of 0.8 mL/min, blood side pressure drops of 60 mmHg and wall shear stresses within physiologic range. A 60-mmHg pressure drop was selected to be compatible with pumpless operation via peripheral arterio-venous pressures [39]. In normal human vasculature, venular wall shear stress ranges from 1–6 dynes∙cm^−2^ (43–260 s^−1^), and arteriole wall shear stress spans 10–70 dynes∙cm^−2^ (434–3038 s^−1^) [7]. Ultimately, the blood-side network of each μAL was designed to contain two distribution channels (input and output) connected by capillaries, achieve equivalent rated flows and pressure drops and fit within 6” diameter circles. Surface area, priming volume and average capillary wall shear were calculated and compared for each capillary height.

Gas exchange and pressure drop calculations were used to determine the number and length of artificial capillaries necessary for each design (Equations (1)–(3)) [5,34,38]. Equations (1) and (2) were first used to determine the total required gas exchange surface area A_c,g_ for the target rated blood flow, where Q_R_ is defined as the highest blood flow rate at which an inlet blood oxygen saturation of 70% can be increased to 95% upon exiting from the outlet [9]. A_c,g_ includes the portion of the capillaries that actively contributes to gas exchange and excludes distribution channels. Once A_c,g_ was determined, the target pressure drop and Equation (3) were used to solve for the number and length of artificial capillaries via Solver, a Microsoft Excel Add-in.
(1)$$Q_{R}=\frac{A_{c,g}}{S_{B,O2}\cdot R_{D,O2}\cdot \ln\;\left(\frac{\mathrm{PO}2_{B,i}-\mathrm{PO}2_{G}}{\mathrm{PO}2_{B,o}-\mathrm{PO}2_{G}}\right)\;},$$

In Equation (1), A_c,g_ is the available capillary gas exchange area, S_B,O2_ is the average effective solubility of oxygen in blood and R_D,O2_ is the membrane’s effective resistance to diffusion. PO2_B,i_, PO2_B,o_ and PO2_G_ are the partial pressures of oxygen in the blood entering and exiting capillaries and in the sweep gas, respectively. In normal human blood, S_B,O2_ is a constant, at approximately 7.9 × 10^−4^ mL-O_2_∙mL-blood^−1∙mmHg−1^ [34,40]. A sweep gas of pure oxygen meant PO2_G_ was defined as 760 mmHg. PO2_B,i_ and PO2_B,o_ were defined as 36.4 mmHg and 79.2 mmHg, corresponding to SO_2_ values of 70% and 95%, respectively. Thus, R_D,O2_ was calculated to solve for A_c,g_.
(2)$$R_{D,O2}=\frac{\delta_{M}}{P_{M,O2}}+\frac{H/2}{S_{B,O2}\cdot D_{B,O2}},$$

$R_{D,O2}$ depends on membrane thickness (δ_M_), channel height (H), membrane permeability to oxygen (P_M,O2_), effective diffusivity of oxygen in blood (D_B,O2_) and S_B,O2_. Devices were designed to contain 30 μm-thick membranes constructed from PDMS, which corresponds to a P_M,O2_ of 3.6 × 10^−7^ mL-O_2_∙cm^−1^∙min^−1^∙mmHg^−1^. The FDA Guidance for Cardiopulmonary Bypass Oxygenators 510(k) Submissions defines normal human blood as having a D_B,O2_ of 1.4 × 10^−6^ cm^2^∙s^−1^ [40].

Pressure drop for capillaries and distribution channels was determined using Equation (3), a combination of Poiseuille’s law and Kirchhoff’s voltage law [41], where Q is flow rate.
(3)$$∆P=\frac{12\cdot \mu \cdot L}{H\cdot W^{3}\cdot \left(1-\frac{0.63\cdot H}{W}\right)}Q,$$

Murray’s law describes how blood vessels scale and branch in the context of natural lungs; it was used here to determine how the distribution channels change in size at each bifurcation (Equations (4) and (5)) [35,36,37]. More specifically, Murray’s law states that a daughter channel with radii r_1_ bifurcates from its respective parent vessel with radii r_p_ at an angle α. r_2_ is the radius of the parent vessel that continues on after the daughter channel branches off (Figure 2). Since the channels used in blood side designs were all rectangular, the various hydraulic radii (r_H_) were converted to channel width (W) and height (H) using $r_{H}=H\cdot W/\left(H+W\right)$.
(4)$$r_{0}^{3}=r_{1}^{3}+r_{2}^{3},$$
(5)$$\cos\;\left(\alpha \right)=\frac{{\left(r_{1}^{3}-r_{2}^{3}\right)}^{4/3}-r_{1}^{4}-r_{2}^{4}}{2r_{1}^{2}r_{2}^{2}},$$

Three designs corresponding to three artificial capillary heights (H_c_ = 30, 60, 100 μm) spanning the mathematically modeled range were selected for blood flow simulations in CFD. After which, the three designs were implemented via soft lithography and testing in vitro to verify performance and CFD simulations.

### 2.3. Computer Aided Drawing and 3D Modeling

DraftSight was used to draw two blood layers (capillary and distribution) and one gas layer for designs with artificial capillary heights of 30, 60, or 100 μm. Capillaries were kept parallel to each other and distribution channels were curved to accommodate branching angles determined via Murray’s law (Equation (5)). Due to the extreme curving of the distribution channels in H_c_ = 30, the first and last 30 capillaries in the design were also curved to allow each capillary to extend their full, assigned length. Corresponding gas side designs were also made in DraftSight. In each design, pillars were 300 μm wide, and the number and spacing of pillars were limited such that pillars occupied 11% of the gas side surface area.

Two-dimensional designs from DraftSight were rendered into three-dimensional models in SolidWorks. Inlet volume flow and outlet pressure boundary conditions were assigned as 0.8 mL/min and 760 mmHg, respectively. Inlet flow was applied uniformly across the inlet distribution channel’s cross-sectional area. Outlet pressure was applied uniformly at the outlet distribution channel’s cross-sectional area.

The Carreau model accounts for shear rate dependent viscosity [42], and was thus used to model blood flow in CFD simulations. Several other studies were used to make slight modifications to blood flow properties [14,26,43,44,45,46]. Table 1 describes the specific blood properties that were used.

### 2.4. Blood and Gas Mold Construction

Blood and gas molds were made from conventional photolithography and soft lithography techniques as follows. First, negative photoresist SU-8 2035 was spin-coated onto 6″ silicon wafers at speeds that would produce an artificial capillary height of 30, 60 or 100 μm. Next, two soft bakes, UV-exposure with a photomask, two post-exposure bakes and a development step were performed to pattern and cure the photoresist. Microchem’s data sheet was used as a starting point for spin speeds, bake temperatures, bake durations and UV-exposure durations [47], but adjustments were necessary to account for our laboratory’s humidity and temperature levels [48]. Table A1 reflects our optimized settings for producing device molds on silicon wafers. A Dino-Lite microscope was used to measure channel heights in triplicate.

To form distribution channels, SU-8 2075 was applied atop the capillary layer by repeating previously described steps. Two layers of SU-8 2075 were necessary to achieve thicknesses greater than 200 µm. Alignment markers were used to ensure capillaries aligned with distribution channels. Specific processing times and settings are outlined in Table A1. Gas channels were formed on a separate wafer using the same procedure used to form the distribution channels.

### 2.5. Device Construction

Devices were constructed using standard soft-lithography techniques, as has been demonstrated in previous works [1,12,14,43,49,50,51,52,53,54]. In short, a 10:1 mixture of Dow Corning Sylgard 184 silicone elastomer base and its curing agent were mixed, degassed in a desiccator for one hour, then poured into blood and gas molds to a thickness of ~7.5 mm. To form diffusion membranes, 10 g aliquots of PDMS were poured onto 200 mm diameter acrylic substrates and spun to a thickness of 30 µm using a Specialty Coating Systems™ G3P-12 spin coater (2550 rpm, 30 s). All three PDMS-carrying vessels were then baked for one hour at 85 °C. Tweezers and razor blades were used to gently separate the cured blood and gas layers from the blood and gas molds and to trim each layer of excess PDMS. 3-mm biopsy punches were used to create perpendicular input and output (I/O) holes in each gas and blood layer. Then, 1-inch long 1/8″ ID silicone tubing connectors were bonded to I/O holes using oxygen plasma (35-watt RF power, 70 mTorr base pressure, 900 mTorr process pressure, 25 s plasma exposure). RTV silicone sealant was applied around the connectors to guarantee fluid-tight seals. Finally, blood, diffusion membrane and gas layers were irreversibly bonded together using the same oxygen plasma settings mentioned above. Completed devices were covered and stored in a non-airtight plastic container at room temperature until used for in vitro testing.

### 2.6. In Vitro Flow Experiments

Initial tests showed incomplete capillary filling in uncoated H_c_ = 30 devices, so H_c_ = 30 devices were re-made and coated with polyethylene glycol (PEG) to ensure gas exchange and pressure drop data would be reflective of complete capillary filling. PEG coating followed our previously established methods [20]. Given that the primary variables of interest (pressure drop and gas exchange) would depend on complete filling of all capillaries, H_c_ = 60 and H_c_ = 100 devices were left uncoated, since their capillary beds perfused easily at all experimental flow rates.

Figure 3 illustrates the experimental layout. The blood bag of whole bovine blood anticoagulated with sodium citrate (1:9) was submerged in a 2–8 °C water bath and gently rocked. A peristaltic pump was used to send blood through a stretch of silastic tubing with an inner diameter (ID) of 1/8″. Blood was warmed to body temperature by directing flow through tubing submerged in a 37 °C water bath. Blood then traveled through a PEG coated preconditioning μAL previously designed in our lab with a rated flow of 20 mL/min [38]. A mixture of nitrogen, oxygen and carbon dioxide was supplied to the preconditioning device in order to achieve target inlet blood gases according to the FDA Guidance for Cardiopulmonary Bypass Oxygenators 510(k) Submissions [40]. Before blood entered the test device, blood was directed through a 1/16″ ID segment of tubing from which inlet pressure and inlet blood gas measurements were taken.

Prior to each experiment, devices were primed with normal saline. Devices were tested at 0.2, 0.6, 0.8 and 1.2 mL/min. An inlet blood gas was taken immediately before each device was tested to ensure blood was being sufficiently preconditioned. After any change in blood flow rate, blood was permitted to flow through the μAL until its blood volume had been replaced at least 2× before taking the next measurement. Experiments were performed in triplicate (3 devices) at blood flow rates of 0.6, 0.8, 1.0 and 1.2 mL/min, with pure oxygen sweep flowing at 4× the blood flow rate.

### 2.7. Statistics

Pressure drop and oxygen saturation data for each design at each flow rate were combined into means and standard deviations. A single factor analysis of variance (one-way ANOVA) test was used to assess for statistically significant differences between results from the three designs at each flow rate. If the one-way ANOVA *p*-value was less than the alpha value (0.05), data was further analyzed to detect which specific pairs of designs were statistically significant. Paired comparisons were made using post-hoc Tukey-Kramer and deemed significant given alpha = 0.05. All statistical analyses were performed in Microsoft Excel.

## 3. Results

### 3.1. Mathematical Modeling for H_c_ = 10 to 100 μm

Previously proven mathematical models were used to design μALs with varying capillary heights, but equal rated flow and pressure drop. Results are shown in Figure 4. Gas exchange surface area increased with capillary height while total (capillaries and distribution channels) blood contacting surface area (A_b,c_) and V_prime_ exhibited a maximum and capillary wall shear rate (τ_w_) exhibited a minimum around H_c_ = 40 μm.

### 3.2. Implementation of 3 Specific Designs (H_c_ = 30, 60, 100 μm)

Based on the results of the mathematical modeling, three designs were chosen for further modeling and implementation. Designs with capillary heights ≤ 25 μm were removed from consideration since their resulting designs would not fit on 6″ silicon wafers. To evenly represent the reduced range, designs with H_c_ of 30, 60 and 100 μm were selected. Resulting channel dimensions for H_c_ = 30, 60 and 100, respectively, are: capillary widths of 120, 240 and 400 µm; capillary lengths of 0.195, 2.001 and 5.610 cm; distribution channel heights of 239, 210 and 241 µm; distribution channel widths between 27–957, 62–841 and 120–964 µm; distribution channel lengths of 9.144, 1.4623 and 0.700 cm. We achieved channel heights within 3% of intended dimensions. For the sake of brevity, individual branching angles are not included; however, branching angles for H_c_ = 30, 60 and 100 ranged from 85.2–54.0°, 79.0–70.9° and 73.7–54.0°, respectively. Implemented branching angles were within 1–12% of what was theorized using Murray’s Law (Equation (5)).

The available gas exchange area was divided across individual capillaries and their corresponding distribution channels. Therefore, H_c_ = 30, 60 and 100 required gas exchange surface areas of 1.2, 2.1 and 3.1 cm^2^, respectively, to meet rated flow, pressure drop and wall shear rate goals (Table 2). Despite A_c,g_ increasing with increasing capillary height, H_c_ = 60 required the smallest V_prime_ and caused the smallest A_b,c_ in mathematical modeling (Table 2). Additionally, all three designs produced τ_w_ within the physiological range, but τ_w_ in H_c_ = 30 and H_c_ = 100 were 43 and 29% less than in H_c_ = 60, respectively. Computer-aided drawings (CAD) of each design are shown below (Figure 5).

### 3.3. CFD Simulations

Initial CFD simulations of devices were run at the rated blood flow (0.8 mL/min) and resulted in a substantial range in pressure drops across the three designs (45.0–71.1 mmHg). Although the original pressure drop goal was 60 mmHg, the H_c_ = 30 μm design achieved 51.5 mmHg, so the pressured drop goal was reduced to 50 mmHg and the H_c_ = 60 and H_c_ = 100 μm designs were adjusted so that all designs would have similar pressure drops (Table 2). By revising the capillary lengths in H_c_ = 60 from 2.4428 to 2.0012 cm and H_c_ = 100 μm design from 7.3879 to 5.6099 cm, the range in pressure drop across the three designs narrowed substantially (51.5–53.8 mmHg). Revised designs were used for all subsequent steps.

Meshing in each design was systematically refined to maximize the precision of results while minimizing simulation time. Optimal mesh levels in SolidWorks were determined by increasing the mesh level stepwise until average velocity, wall shear stress and total pressure drop changed by <1%. A Level 3 global mesh was appropriate for H_c_ = 100, but a Level 4 global mesh was necessary in H_c_ = 60 and H_c_ = 30 to capture shear rate details in capillary height-width cross-sections. A Level 4 local mesh provided no more insight than Level 5 in all designs. Ultimately, this meant computational domains for H_c_ = 30, 60 and 100 had 845,767, 1,226,957 and 1,723,605 fluid cells, respectively.

Final simulation results confirmed that each design produced physiologic pressure drops and capillary wall shear rates (Table 2, Figure 6 and Figure 7). Wall shear in capillaries was 1500, 2588 and 1729 s^−1^ in H_c_ = 30, 60 and 100, respectively (Figure 8). In planar Poiseuille flow, shear rate is highest at capillary walls and approaches zero at the center of capillary lumens [55], so maximum shear rates in cross-sections should be equivalent to capillary wall shear rate (as validated in Figure 8, below). Despite achieving physiologic wall shear rates that matched mathematical modeling results (Figure 4), the cross-sections of individual capillaries revealed better symmetry in H_c_ = 60 and H_c_ = 100 than in H_c_ = 30 (Figure 8). Reasons for these differences are discussed in 4.3.

### 3.4. In Vitro Experimental Results

All three implemented designs are shown in Figure 5 filled with blood.

Benchtop experiments to measure experimental pressure drop and gas exchange were run at 37 °C using bovine whole blood with 40% hematocrit (Figure 9). CFD results were within 15% of pressure drops predicted by mathematical modeling, and experimental pressure drops in H_c_ = 30, 60 and 100 devices were 47%, 27% and 32% lower than theory.

One-way ANOVA of pressure drop data indicated a significant difference between one or more design types at each flow rate. *p*-values between the three designs at 0.6, 0.8, 1.0 and 1.2 mL/min were 0.0025, 0.0035, 0.00431 and 0.0101, respectively. Post-hoc Tukey-Kramer tests confirmed that pressure drop was significantly different between H_c_ = 30 vs. H_c_ = 60 at all flow rates, and between H_c_ = 30 vs. H_c_ = 100 at flow rates except 1.2 mL/min (Figure 10). This may be due to the fact that the H_c_ = 30 design was coated with PEG (see Discussion below).

Gas exchange tests were completed using pure oxygen as the sweep gas, and all three designs met or exceeded theoretical expectations for oxygen transfer. There were no statistically significant differences between each design’s ability to oxygenate blood at the three highest flow rates. Change in oxygen saturation was notably high for H_c_ = 100 at 0.6 mL/min. The one-way ANOVA test for comparison at 0.6 mL/min resulted in *p* = 0.032; a post-hoc Tukey-Kramer test was performed, revealing significant differences between only H_c_ = 60 and H_c_ = 100 (Figure 11). Observing similar gas exchange for all three designs validated theoretical modeling and design goals.

## 4. Discussion

### 4.1. Mathematical Modeling for H_c_ = 10 to 100 µm

Fixing rated blood flow meant all three devices had equivalent gas exchange capabilities. By confining designs to share equal pressure drops, we could then directly elucidate the relationship between capillary height and μAL properties. For very small capillary heights, gas exchange is very efficient, necessitating a small gas exchange area to achieve the desired rated blood flow. To meet pressure drop specifications, however, an abundant number of short capillaries must run in parallel, requiring long blood distribution channels to interconnect the numerous capillaries. Long distribution channels have large priming volumes and blood contacting surfaces resulting in large total values for these quantities in Table 2. As capillary height increases, gas exchange efficiency decreases, increasing the required gas exchange area; however, a smaller number of capillaries are required in parallel to meet the pressure drop specification (due to decreased resistance of each capillary). The result is that as capillary height increases, the required blood distribution channel length decreases, decreasing total blood contacting surface area and priming volume. A minimum value for total surface area and priming volume is reached when capillaries are ~42 µm tall, above which the inefficiency of gas exchange in large capillaries necessitates a large capillary gas exchange area, increasing both total surface area and blood priming volume.

Regarding the three specific designs chosen for implementation, H_c_ = 30 requires hundreds of short capillaries, the overall design is long and narrow; H_c_ = 60 requires a few dozen capillaries, so the design resembles a square; H_c_ = 100 contains approximately one dozen long capillaries and looks like a short rectangle (Figure 5). All three overarching design goals were mathematically achieved, as demonstrated in Table 2.

### 4.2. CFD Simulations

The overall shear rate in a majority of H_c_ = 30’s capillaries appear blue (<1041 s^−1^), as is similarly notable in H_c_ = 100’s capillaries, while H_c_ = 60’s capillaries appear green (1041–2039 s^−1^) (Figure 7). This agrees with the mathematical modeling which shows that wall shear rate approaches a maximum when capillaries are 40–60 µm tall (Figure 4, Table 2).

Despite systematically selecting Level 4 for local meshing to ensure sufficient refinement (<1% change in shear rate) and achieving the expected parabolic shear pattern in all three designs, H_c_ = 30’s shear rate cross-section appears less symmetric than the two other designs (Figure 8); this is potentially due to artifacts from low resolution. Although efforts were made to increase the local mesh refinement in H_c_ = 30 from Level 4 to Level 5, computation time became excessive, such that higher-resolution results could not be collected. Furthermore, capillary wall shear simulated via CFD was only 3% higher than predicted by mathematical modeling, so we did not pursue higher refinement.

### 4.3. In Vitro Results

The lower experimental pressure drop observed in H_c_ = 30 may be due to the increased wettability caused by PEG coating. We have found in previous work that a hydrophilic surface reduces pressure drop in small diameter flow channels [20]. Future experiments should control for surface hydrophilicity by applying the same surface modification, such as PEG, to all devices regardless of channel height. Furthermore, CFD simulations assumed devices were constructed from a non-deformable solid, whereas PDMS is inherently flexible, so this may have contributed to in vitro pressure drops being lower than predicted [56]. In vitro pressure drops in H_c_ = 60 and H_c_ = 100 were not significantly different (Figure 10), despite both experimental pressure drops being less than theory. This suggests that the mathematical modeling and CFD of pressure (Table 2) were accurate in predicting similar pressure drops across designs containing differing capillary heights.

Oxygen exchange performance in all designs was very similar at all flow rates. The only significant difference was observed between H_c_ = 60 and H_c_ = 100 at 0.6 mL/min (Figure 11). Change in SO_2_ generally decreased with increasing flow rates, a trend that agrees with other previously reported microfluidic artificial lungs [22,38]. Overall, O2 exchange in test devices met or exceeded theoretical expectations.

### 4.4. Limitations

Despite using alignment markers, we suspect that perfect alignment between capillaries and distribution channels was not achieved in the H_c_ = 30 µm design molds. Before committing a mold for use in device fabrication, preliminary perfusion tests were performed with dyed water. Perfusion tests revealed that although a vast majority of capillaries perfused immediately, flow through H_c_ = 30’s first and last few dozen capillaries was delayed. This delay in perfusion may be due to partially obstructed access between capillaries and distribution channels, a potential result of slight misalignment. Or, as discussed previously, it may have been due to the first/last few dozen capillaries having branching angles that varied from Murray’s law.

Additionally, using two fixed channel heights is a significant simplification of the variable vessel diameters found in natural lungs. This means blood inevitably experienced abrupt transitions in channel height, instead of smooth transitions when flowing into and out of artificial capillaries within our µALs. Similarly, photolithography limits fabrication to rectangular channels, whereas physiologic vessels are round, thus removing these devices a step from true biomimicry.

### 4.5. Continuing Efforts

While artificial oxygenators to date have focused on achieving high gas exchange efficiency and biocompatibility, this work performed a parametric analysis to determine the impact of channel dimension on µAL properties and performance. Notably, capillary heights between 40 to 60 µm minimized blood contacting surface area and priming volume while maximizing wall shear. In vitro testing verified a similar gas exchange and pressure drop between the three implemented designs. This data suggests that future work towards the development of PDMS-based, clinically relevant artificial lungs should consider the role capillary dimensions have in device properties (priming volume, surface area, wall shear rate) given fixed parameters; however, further validation with in vitro tests using human blood would help elucidate the implications of capillary height on biocompatibility.

## 5. Conclusions

In this work, three μALs were developed with differing channel dimensions but equivalent, physiologic parameters, such as rated flow and pressure drop. In CFD simulations, the design containing 60 µm-tall capillaries required the lowest total blood contacting surface area and priming volume but subjected blood to the highest wall shear rate. In vitro experiments using bovine whole blood revealed that gas exchange was similar between all designs, but the two designs with larger capillaries (60 and 100 µm-tall) consistently produced higher pressure drops than the design with smaller capillaries (30 µm-tall), although this may have been due to the hydrophilic surface coating on the design containing 30 µm tall capillaries. Overall, results regarding capillary size and performance may be valuable for guiding future work which manipulates blood flow networks to improve biocompatibility of μALs.

## Figures and Tables

**Figure 1 micromachines-13-00822-f001:**
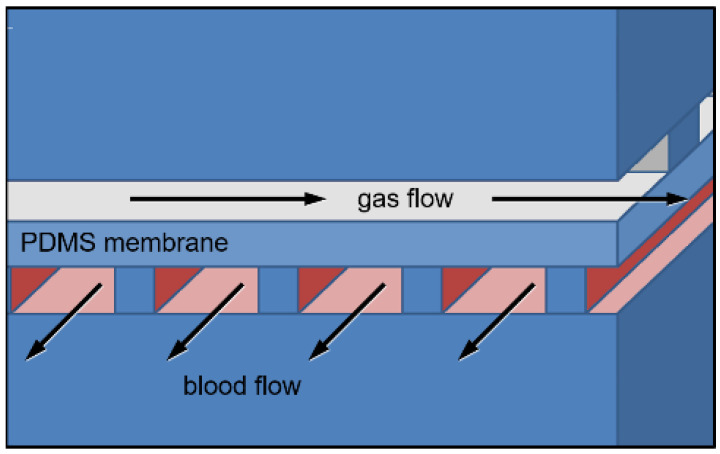
A cross-section drawing of a microfluidic artificial lung. Blood flows in the Z direction and gas flows in the X direction. Image reused with permission from Thomson et al., 2019 [38].

**Figure 2 micromachines-13-00822-f002:**
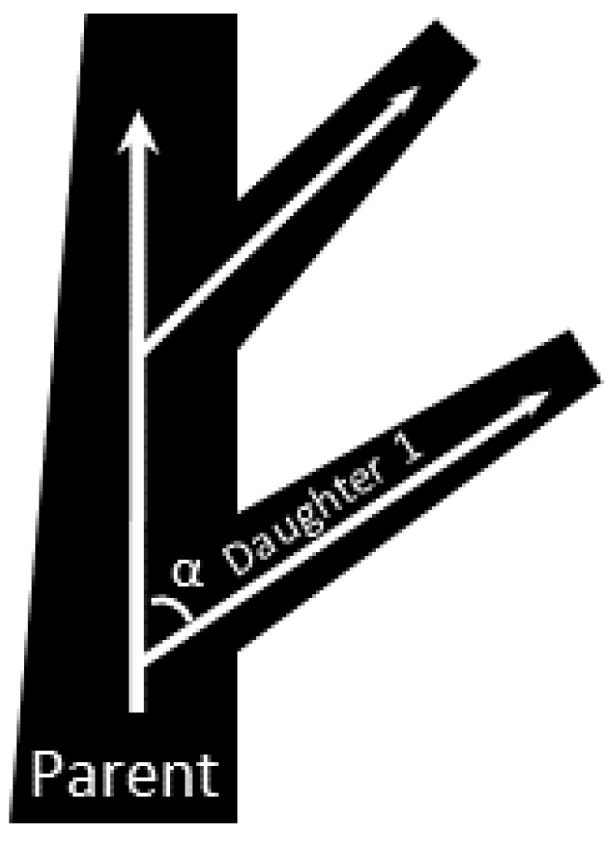
Drawing of daughter channels branching from their respective parent channel.

**Figure 3 micromachines-13-00822-f003:**
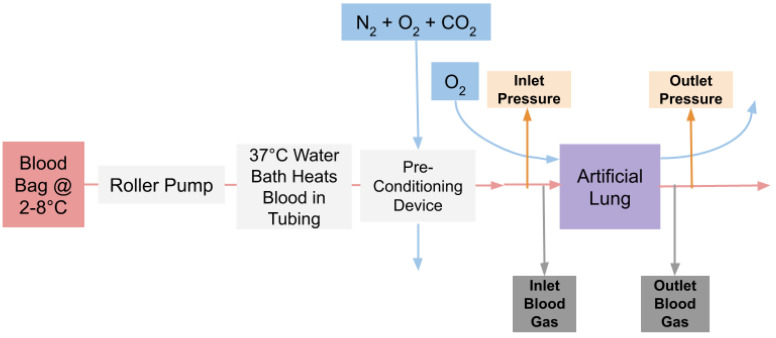
In Vitro circuit set–up.

**Figure 4 micromachines-13-00822-f004:**
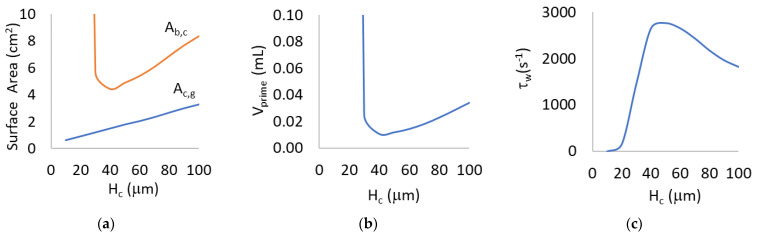
(**a**) Gas exchange surface area by capillaries (A_c,g_), total blood contacting surface area (A_b,c_); (**b**) priming volume in capillaries and distribution channels (V_prime_); and (**c**) wall shear rate in capillaries (τ_w_) with respect to capillary height (H_c_).

**Figure 5 micromachines-13-00822-f005:**
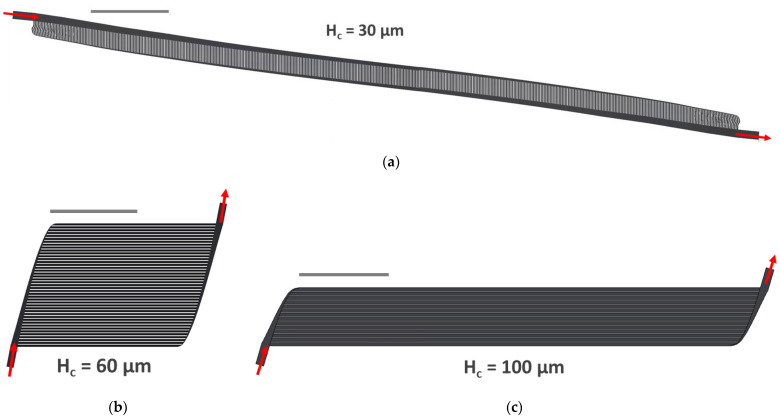
CAD drawing of (**a**) H_c_ = 30; (**b**) H_c_ = 60; and (**c**) H_c_ = 100. Scale bars = 1 cm.

**Figure 6 micromachines-13-00822-f006:**
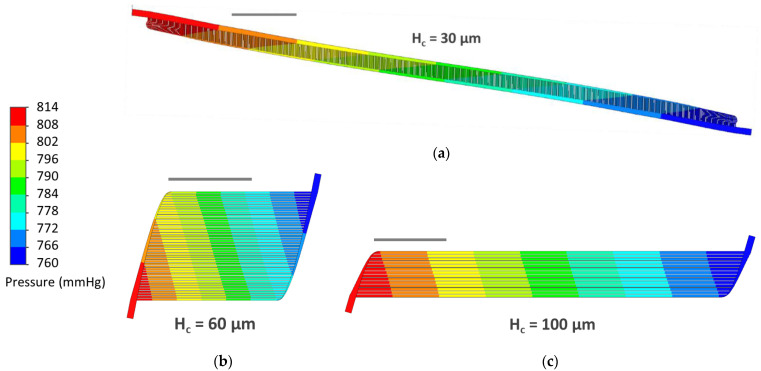
(**a**) Pressure drop cut plots half-way through capillary heights of H_c_ = 30; (**b**) H_c_ = 60; and (**c**) H_c_ = 100. Dark blue corresponds to lower pressures (760 mmHg) and red corresponds to higher pressures (814 mmHg). Scale bars = 1 cm.

**Figure 7 micromachines-13-00822-f007:**
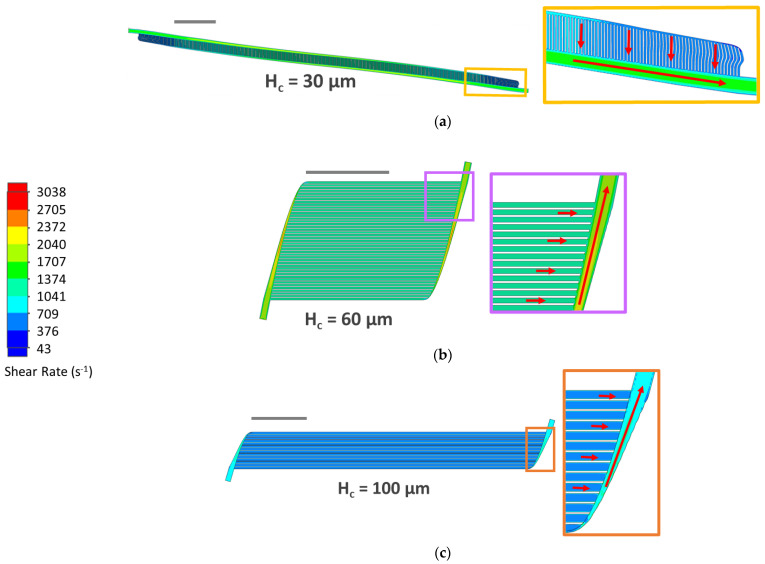
(**a**) Shear rate cut plots halfway through capillary heights of H_c_ = 30; (**b**) H_c_ = 60; and (**c**) H_c_ = 100. Dark blue and red corresponds to the lower and upper bounds of physiological shear rate, 43 and 3038 s^−1^ (1 and 70 dyn·cm^2^), respectively. Scale bars = 1 cm.

**Figure 8 micromachines-13-00822-f008:**
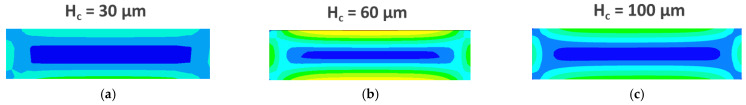
(**a**) Shear rate cut plots at the midpoint of the centermost capillaries in of H_c_ = 30; (**b**) H_c_ = 60; and (**c**) H_c_ = 100. Local mesh Level 4 was applied to maximize resolution. Dark blue and red corresponds to the lower and upper bounds of physiological shear rate, 43 and 3038 s^−1^ (1 and 70 dyn·cm^2^), respectively.

**Figure 9 micromachines-13-00822-f009:**
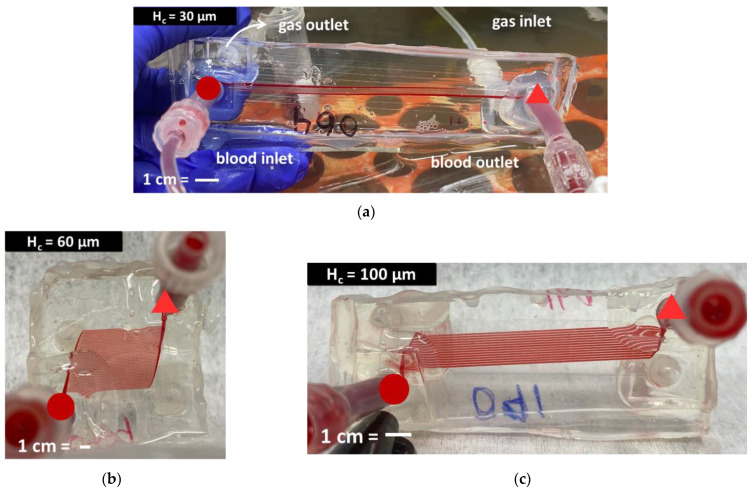
(**a**) H_c_ = 30; (**b**) H_c_ = 60; and (**c**) H_c_ = 100 perfused with blood. All images are oriented such that the inlet is positioned at the bottom left (dark circle) and the outlet and the top right (bright triangle). Scale bars = 1 cm.

**Figure 10 micromachines-13-00822-f010:**
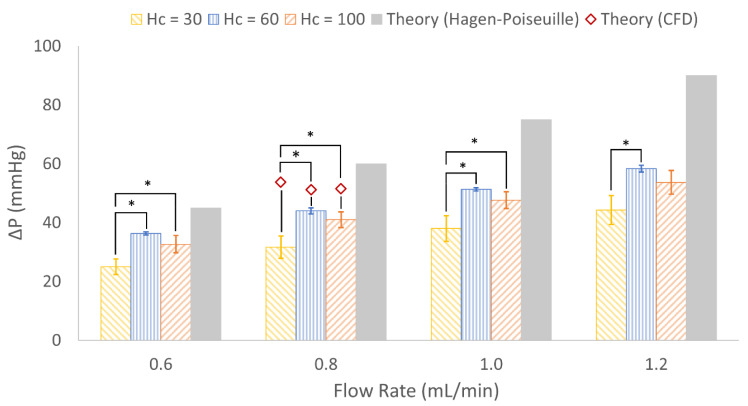
Experimental and computational pressure drops for each design at various flow rates (*n* = 3) when tested with bovine whole blood. At each flow rate, bars represent the following from left to right: H_c_ = 30, H_c_ = 60, H_c_ = 100 and mathematical theory. Red diamonds represent theoretical CFD results. Error bars = mean ± standard deviation. * = statistically significant at alpha of 0.05.

**Figure 11 micromachines-13-00822-f011:**
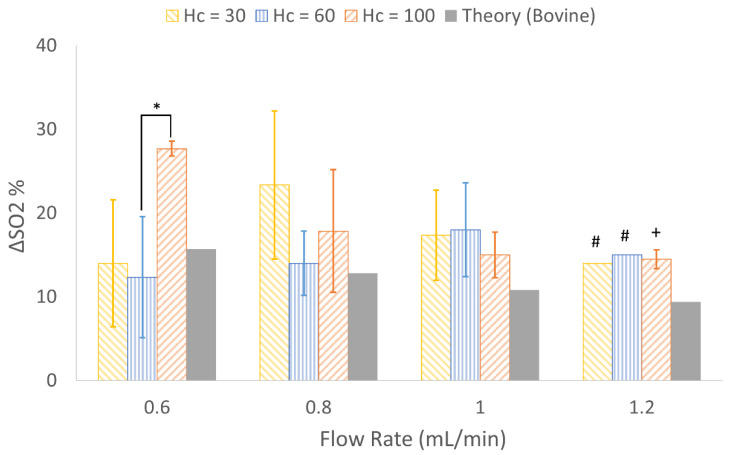
Experimental and theoretical change in oxygen saturation achieved by each design at various flow rates (*n* = 3) when tested with bovine whole blood. At each flow rate, bars represent the following from left to right: H_c_ = 30, H_c_ = 60, H_c_ = 100 and mathematical theory. Error bars = mean ± standard deviation. + n = 1, # n = 2. * = statistically significant at alpha of 0.05.

**Table 1 micromachines-13-00822-t001:** Carreau model blood properties used in CFD simulations.

Blood Density (kg∙m^−3^)	Maximum Dynamic Viscosity (Pa∙s)	Minimum Dynamic Viscosity (Pa∙s)	Power-Law Index	Time Constant (s)
1060	0.056	0.00345	0.3568	3.313

**Table 2 micromachines-13-00822-t002:** Mathematical modeling results (gas exchange surface area of capillaries (A_c,g_), blood contacting surface area (A_b,c_), priming volume in capillaries and distribution channels (V_prime_) and capillary wall shear rate (τ_w_) of H_c_ = 30, 60 and 100 at rated flow (Q_R_) compared to pressure drop (ΔP) and τ*_w_* from CFD simulations.

		Mathematical Theory					CFD	
Designs	Q_R_ (mL/min)	A_c,g_ (cm^2^)	A_b,g_ (cm^2^)	V_prime_ (µL)	ΔP (mmHg)	τ_w_ (s^−1^)	ΔP (mmHg)	τ_w_ (s^−1^)
H_c_ = 30	0.8	1.2	5.4	25	60	1458	53.8	1500
H_c_ = 60	0.8	2.1	5.0	15	60	2546	51.2	2588
H_c_ = 100	0.8	3.1	7.3	33	60	1818	51.5	1729

## Data Availability

Not applicable.

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
