# Peer review of "A Parametric Analysis of Capillary Height in Single-Layer, Small-Scale Microfluidic Artificial Lungs"

_micromachines, 2022, doi:10.3390/mi13060822_

Round 1
Reviewer 1 Report
The authors mainly reported a parametric analysis for optimizing the capillary height in single layer, small scale microfluidic artificial lungs (μALs). By using CFD (based on Murray’s law and the Hagen-Poiseuille equation) simulations and in-vitro experiments, the authors evaluated different designs with capillary height 30, 60, and 100 μm under fixed blood flow and total pressure drop conditions. According to the authors, this analysis may be valuable for guiding μALs design in order to optimize their performances and overcome biocompatibility-related issues, such as non-physiologic flow pattern and clotting or hemorrhagic complication risks. In general, this parametric analysis is thorough and meaningful. However, there are also some problems suggested to be addressed before publication.
Comment 1:
In 2.6 In vitro Flow Experiments section, is the preconditioning device similar to the μAL or a different one? Please give more detailed description about how the preconditioning device is designed and how it works.
Comment 2:
In 2.6 In vitro Flow Experiments section, the authors described that Hc = 30 devices were coated with PEG to address incomplete capillary filling. Why not add this process for Hc = 60 and 100 μm devices to ensure experimental consistency?
Comment 3:
In 3.2 Implementation of 3 Specific Designs (Hc = 30, 60, 100 μm) section, what are dimensions of designed gas layer channels? Is there any design rule about distance between each channel? Please give more descriptions.
Comment 4:
In 3.1 Mathematical Modeling for Hc = 10 to 100 μm section, mathematical modelling results indicate that the μAL has best performance with Hc ≈ 40 μm. Why not add design with Hc = 40 μm in the following experiment to evaluate the extreme value?
Comment 5:
In 3.4 In Vitro Experimental Results section, experimental pressure drops in Hc = 30, 60, and 100 devices showed different degrees of decrease relative to the theoretical values. Please describe and discussion the reason of those different degrees of decreases.
Reviewer 2 Report
This is an interesting parametric study that attempts to isolate the effect of microchannel height on efficiency and run requirements for microfluidic artificial lung systems (uAL). The combination of mathematical modeling, finite-element CFD, and experimental verification is particularly nice, and the overall suggestion that smaller is not necessarily better (for channel height) is thought-provoking. I have a few comments:
- What is the motivation for selecting 60 mm Hg as the blood-side pressure drop? If this is simply a convenient choice for standardization below a maximum practical value, please say that. At a first glance it seems that the choice of pressure drop determines what microchannel height leads to a square uAL – if this is the case it is probably worth a sentence in the discussion.
- Please make explicit whether or not the mathematical theory as described in text and in the middle columns of Table 2 includes the contribution of the distribution channels. I assume V_prime includes the distribution channels (?), but I am not sure about Acg and Tw.
- I find some aspects of the presented calculations and discussion around shear to be confusing. In particular:
- 1. It seems to me that the important physiological comparison is the shear rate at/near the vessel wall; for Poiseuille flow, shear will always be minimal in the middle of the channel. It is not clear to me whether the design parameter being considered for the microchannels is wall shear rate, or average shear rate, which will be considerably lower (as an aside, please check whether ref2 on page 3 line 103 is the correct one). For example:
“Under normal physiological flow conditions, the wall shear rate increases from about 10 s-1 in veins to about 2000 s-1 in the smallest arteries, whereas maximal wall shear rates up to 40,000 s-1 have been described for severe atherosclerotic arteries “ https://www.ncbi.nlm.nih.gov/pmc/articles/PMC5137878/ (I am definitely not insisting the authors cite this particular paper - I am just choosing this one to point out the mention of -wall- shear rate because it is available open access on pubmed).
2. Along those lines, I don’t understand the decision to show the shear rate cut plots in Fig. 7 at the shear rate midpoint (?), rather than at a fixed (small) distance from the channel wall for each geometry. Furthermore, if there is a concern about exceeding physiological levels of wall shear stress, it seems like the distribution channels in Fig.7b sliced more closely to the device surface might be close (which could likely be resolved in future work by modifying the distribution channel design if needed).
3. If the blood is being treated as a homogeneous fluid, I cannot understand the shear rate cross sections shown in Fig. 8. Unless there’s an artifact from meshing, these profiles should be similar in shape between the channel sizes. It’s possible that the 30um height data (Fig8a) will show this in line profiles, and that the apparently uniform shear is just a function of low value resolution on the colormap, but the one color boundary shown is not top-to-bottom symmetric. Could the authors please check some scaled line plots to make sure their models are running correctly?
4. In particular, this sentence, p 8 L 293: “Specifically, shear in capillaries ranged from 293 823-1394, 241-2419, 104-1682 s-1 in Hc = 30, 60, and 100, respectively.” does not really make sense. In Poiseuille flow (again, assuming the CFD is treating blood as a homogeneous fluid), the velocity profile across a channel is parabolic, which means that the shear in the middle of the channel is zero - and any non-zero value you calculate or measure is an artifact of your spatial resolution. Please focus on wall shear rate. The discussion p.12 L 367-383 also needs to be edited accordingly.
- Could the authors please comment on whether the 30um measurements might be showing some effects from the presence of RBCs, particularly whether there’s any packing or variations in RBC density occurring near the transition from the distribution channels to the capillaries.
Minor corrections:
- Kayaku not Kayuka
- 11 is missing diamonds from CFD calculations
- 4 line 160 please specify what is meant by “inner face of the inlet distribution channel”
Round 2
Reviewer 1 Report
The authors addressed most of my concerns, and I suggest the paper can be accepted in its current form.